# Responses of the putative trachoma vector, *Musca sorbens*, to volatile semiochemicals from human faeces

Ailie Robinson[1]*, Julie Bristow[1,2¤a], Matthew V. Holl[1¤b], Pateh Makalo[3¤c], Wondu Alemayehu[4], Robin L. Bailey[5], David Macleod[6], Michael A. Birkett[2¤d], John C. Caulfield[2¤d], Virginia Sarah[7], John A. Pickett[2¤e], Sarah Dewhirst[8], Vanessa Chen-Hussey[8], Christine M. Woodcock[2¤d], Umberto D'Alessandro[3¤c], Anna Last[5], Matthew J. Burton[5], Steve W. Lindsay[9], James G. Logan[1]

**1** Department of Disease Control, London School of Hygiene and Tropical Medicine, Keppel Street, London, United Kingdom, **2** Biological Chemistry and Crop Protection, Rothamsted Research, Harpenden, Hertfordshire, United Kingdom, **3** Medical Research Council Unit, The Gambia, The Gambia, **4** The Fred Hollows Foundation, Addis Ababa, Ethiopia, **5** Department of Clinical Research, London School of Hygiene and Tropical Medicine, London, United Kingdom, **6** Department of Infectious Disease Epidemiology, London School of Hygiene and Tropical Medicine, London, United Kingdom, **7** Global Partnerships Executive, The Fred Hollows Foundation, Crawford Mews, London, United Kingdom, **8** ARCTEC, Chariot Innovations Ltd, London School of Hygiene and Tropical Medicine, London, United Kingdom, **9** Department of Biosciences, Durham University, Durham, County Durham, United Kingdom

¤a Current address: The University of Sussex, Falmer, Brighton, United Kingdom
¤b Current address: Entocycle Ltd., United Kingdom
¤c Current address: Disease Control and Elimination, Medical Research Council Unit, The Gambia at the London School of Hygiene and Tropical Medicine, Fajara, The Gambia
¤d Current address: Biointeractions and Crop Protection, Rothamsted Research, Harpenden, Hertfordshire, United Kingdom
¤e Current address: School of Chemistry, Cardiff University, Cardiff, United Kingdom
* Ailie.robinson@lshtm.ac.uk

**Data Availability Statement:** All relevant data are within the manuscript and its Supporting Information files.

## Abstract

The putative vector of trachoma, *Musca sorbens*, prefers to lay its eggs on human faeces on the ground. This study sought to determine whether *M. sorbens* females were attracted to volatile odours from human faeces in preference to odours from the faeces of other animals, and to determine whether specific volatile semiochemicals mediate selection of the faeces. Traps baited with the faeces of humans and local domestic animals were used to catch flies at two trachoma-endemic locations in The Gambia and one in Ethiopia. At all locations, traps baited with faeces caught more female *M. sorbens* than control traps baited with soil, and human faeces was the most successful bait compared with soil (mean rate ratios 44.40, 61.40, 10.50 [*P*<0.001]; 8.17 for child faeces [*P* = 0.004]). Odours from human faeces were sampled by air entrainment, then extracts of the volatiles were tested by coupled gas chromatography-electroantennography with laboratory-reared female *M. sorbens*. Twelve compounds were electrophysiologically active and tentatively identified by coupled mass spectrometry-gas chromatography, these included cresol, indole, 2-methylpropanoic acid, butanoic acid, pentanoic acid and hexanoic acid. It is possible that some of these volatiles govern the strong attraction of *M. sorbens* flies to human faeces. If so, a synthetic blend of these chemicals, at the correct ratios, may prove to be a highly attractive lure. This could be

**Funding:** This work was supported by (1) a studentship to JB jointly funded by the Biotechnology and Biological Sciences Research Council and the Medical Research Council; (2) The Fred Hollows Foundation; and (3) The Wellcome Trust through the Stronger SAFE Collaborative Award to MJB, VS and JGL (206275/Z/17/Z). The Fred Hollows Foundation applied untied funds to this project and staff participated in the study design; in the collection, analysis, and interpretation of data; in the writing of the report; and in the decision to submit the paper for publication. The Wellcome Trust had no involvement in the study design; in the collection, analysis, and interpretation of data; in the writing of the report; and in the decision to submit the paper for publication.

**Competing interests:** The authors have declared that no competing interests exist.

used in odour-baited traps for monitoring or control of this species in trachoma-endemic regions.

## Author summary

*Musca sorbens*, also known as the Bazaar Fly, visits people's faces to feed on ocular and nasal discharge. While feeding, *M. sorbens* can transmit *Chlamydia trachomatis*, the bacterium that causes the infectious eye disease trachoma. Around 1.9 million people worldwide are visually impaired or blind from this disease. Although it is believed that *M. sorbens* transmits trachoma, very few studies have looked at ways to control this fly. A large-scale trial has shown that control of fly populations with insecticide reduces active trachoma disease prevalence. Odour-baited traps for the suppression of disease vector populations are an attractive option as there is no widespread spraying of insecticide; however, highly attractive baits are critical to their success. Here, we demonstrate that the preference of these flies for breeding in human faeces is probably mediated by odour cues, and we isolate chemicals in the odour of human faeces that cause a response in the antennae of *M. sorbens*. These compounds may play a role in the specific attractiveness of human faeces to these flies, perhaps by being present in greater amounts or at favourable ratios. These may be developed into a chemical lure for odour-baited trapping to suppress *M. sorbens* populations.

## Introduction

The Bazaar Fly, *Musca sorbens*, is the putative vector of the blinding eye disease trachoma [1]. Adult *M. sorbens* feed on ocular and nasal secretions to obtain nutrition and liquid [2], and in doing so can transmit *Chlamydia trachomatis*, the bacterium that causes trachoma, from person to person. *Chlamydia trachomatis* DNA has been found on wild caught *M. sorbens* [3–5], and a laboratory study demonstrated mechanical transmission of *Chlamydia psittaci* between the eyes of guinea pigs by the closely related *Musca domestica* [6]. Strong evidence for the role of *M. sorbens* as a vector of trachoma comes from a cluster-randomised controlled trial that examined the impact of fly control interventions on trachoma prevalence [7]. Insecticide spraying significantly reduced the number of *M. sorbens* flies caught from children's faces by 88%, and a 56% reduction in trachoma prevalence in children was observed. The provision of pit latrines, which by removing sources of open defecation controls *M. sorbens* juvenile stages, resulted in a 30% decrease in flies on faces and a 30% reduction in trachoma prevalence (non-significant).

These findings demonstrate that controlling the population density of *M. sorbens* may contribute to a decline in trachoma by reducing the number of fly-eye contacts, highlighting the disease-control potential of effective fly control tools. Odour-baited traps are receiving increased attention with regards to disease vectors, as the knowledge base around insect olfaction and attractive volatile chemicals expands [8–12]. Recent studies demonstrating the epidemiological significance of implementing chemical-based lures for vector-borne disease control [13] bolster the more widely accepted, and longstanding, tsetse fly example [14].

Mass deployment of an odour-baited trap for *M. sorbens*, based on the attractive volatiles in faeces, may suppress populations sufficiently to decrease the prevalence of trachoma.

Alternatively, such traps could be used for entomological surveillance, and for monitoring and evaluating *M. sorbens* control programmes.

Female *M. sorbens* deposit their eggs on faeces, in which the larvae develop [15]. Previous studies have shown that *M. sorbens* preferentially oviposit in human faeces [2,16], and that adults emerging from human faeces are on average larger than those emerging from non-human faeces. This suggests that human faeces may be an optimal larval development medium [16]. More prolific emergence from human faeces could be a result of better larval survival within human faeces, and/or more oviposition in human faeces caused by its relatively greater attraction to female flies, or other unknown factors. It is common for insects to use semio-chemicals, volatile airborne chemical signals, to locate resources such as oviposition sites, and to discriminate between resources of varying quality. It is therefore plausible that female *M. sorbens* visit human faeces more frequently relative to the faeces of other animals because of favourable semiochemical cues.

The aim of this study was to investigate the relative attractiveness of faeces from different mammalian species, including humans and local domesticated animals, to the Bazaar fly *M. sorbens* at two locations in The Gambia and one in Ethiopia. Using chemical ecology tech-niques, we then sought to identify putative *M. sorben* attractants in human faecal odours.

## Methods

### Study areas and study periods

The study was carried out in The Gambia and Ethiopia. In The Gambia, there were two sites, the village of Boiram, Fulado West, Central River Division, and the rural town of Farafenni. Studies were conducted in Boiram during the June-August 2009 rainy season, and in Farafenni in November-December 2009, immediately after the rainy season. The third site was Bofa Kebele (Oromia, central Ethiopia), identified as having highly prevalent active trachoma by the Global Trachoma Mapping Project. There, the study was conducted in February 2017 [17].

### Ethics

The Gambian study was approved by the Joint Gambian Government/Medical Research Council Laboratories Joint Ethics Committee (protocol number L2010.90, re: L2009.67, 01/12/09), and the Ethiopian study by the LSHTM ethics committee (reference number 11979/RR/5821) and the Oromia Regional Health Bureau Ethics Committee. In the Gambian study, writ-ten informed consent was provided by the heads of all compounds in which traps were sited or from which faeces were collected. In the Ethiopian study, written informed consent was pro-vided by all participants including guardians of children.

### Trapping

**Trap design.** Fly traps were placed on the ground. The basic design of the 'Bristow Trap' was a lidded white plastic pot (Vegware, Edinburgh, UK) containing bait, with bait volatiles being released through a hole in the lid covered with mesh (Fig 1). Baits were 50 g of faeces, soil, or none (empty). The pot lid was made sticky to catch flies, by attaching circles cut from 'sticky paper' (Gambian study, "Yellow Sticky Trap" [Agrisense BCS Ltd, Pontypridd, UK]; Ethiopian study, "Yellow Sticky Trap" [Agralan, Wiltshire, UK]. A nylon (Gambian study, 0.4 mm gauge [Lockertex, Warrington]) or polyester (Ethiopian study, unknown gauge size [The Textile House, Huddersfield]) mesh was used to cover the hole to prevent fly entry. Traps were replaced daily, and a barrier of thorny acacia branches was placed in front of all traps to stop animals interfering with them.

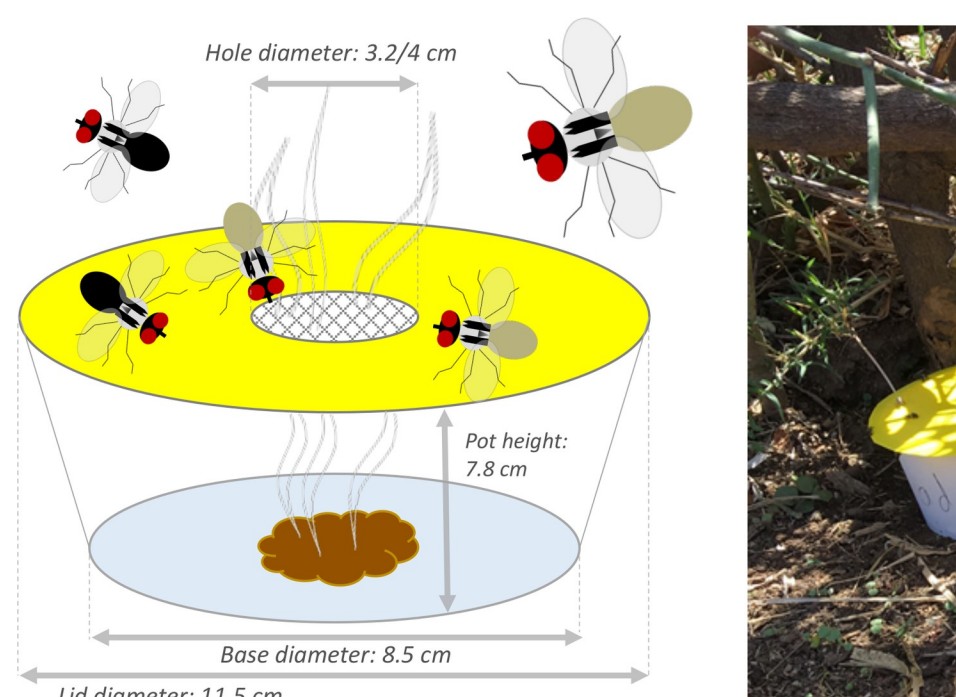
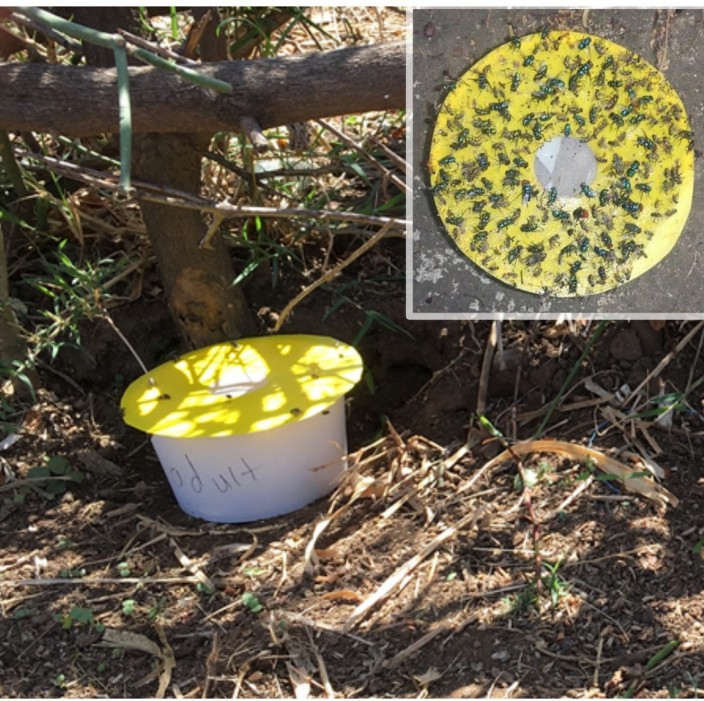

**Fig 1. 'Bristow Trap'—an odour-baited trap for *Musca sorbens*.** (A) Trap design and dimensions (hole diameter used in the two studies was different; Gambian study 3.2 cm, Ethiopian study 4 cm) (B) A fly trap 'in situ' in Ethiopia; (insert) typical fly catch on the yellow sticky discs on the trap.

In the Gambian studies only, the following modifications were used. A layer of glue ("Rat Stop", 92% polybutene, 8% hexane [Taizhou Aiyon Adhesive Co., China]) was added to the top of the yellow disc to increase trap catch. A thin band of the same glue was applied to the side of each pot to prevent ants from gaining access to trap catch. To protect from rain, traps were placed inside wire frame boxes (15 cm x 15 cm x 15 cm [locally sourced]) with blue plastic sheeting (locally sourced) on the uppermost side and above the trap.

**Collection of faeces bait.**   In the Gambian study, faeces (50 g; calf, cow, dog, donkey, horse, human and sheep) were collected for trap bait from open defecation in the compounds between 07:00 h and 11:30 h and weighed on a balance (Salter ARC 1066, accurate to ± 1 g). In Boiram, human faeces were obtained from two adjacent compounds, the children of which defaecated on the ground. In Farafenni, human faeces were obtained from a compound with five children between the ages of eight and 15 years, who defaecated in a plastic potty in the morning. Only fresh faeces (< 1 day old) were used in traps. Cow and calf faeces were collected from areas where cattle were confined at night. Cattle grazed in the bush, calves were under one-year-old and fed on milk. Dog faeces were unavailable in Boiram. Donkeys and horses were fed cous (millet), supplemented in the village by grazing. Their faeces were collected from family compounds. Sheep faeces were collected from shelters where the animals were kept overnight. Sheep in the village grazed in the bush, whilst those in town scavenged in the streets. Tobaski, a religious festival during which rams or sheep are slaughtered, occurred between the first and second experiments. Before Tobaski, sheep had their diets supplemented with milk, millet porridge or Senegalese feed blocks known as *repas* of unknown composition. After Tobaski the sheep had a less nutritious diet. Soil samples for control bait were taken from areas close to the trap site, but ensuring that no faeces were in the immediate vicinity (within 5 m). In the Ethiopian study, human adult and human child, cow and donkey faeces

(50 g) were collected from the compounds where open defecation is commonplace, and weighed (Ascher Portable Digital Scale, accurate to 0.01 g).

**Experimental design and data analysis.**   Traps containing faeces or soil bait were set daily, 50 cm apart, along a transect. Latin Square (LS) designs were used so that baits were rotated between trap positions daily, to prevent trapsite variation and bias, or adjacent trap bait, have any effect on trap catch. In the Gambian study, eight traps were set daily (seven faeces baits and a soil, with an empty pot instead of dog faeces in Boiram). The eight by eight LS was repeated twice at each site, giving 16 trap days per location. In the Ethiopian study, a transect of five traps (four faeces and one soil bait) was set on day one outside one household, and thereafter five-trap transects were deployed outside two (new) households daily until day six, giving a total of 11 trapping days per treatment (bait type). The position of the traps within each transect was rotated according to a LS design. Environmental variables were recorded as follows: Bioram, max/min ambient temperature, presence/absence of rain (thermometers hung nearby and daily observation of rain); Farafenni max/min ambient temperature and humidity (TinytagPlus datalogger, Gemini dataloggers); Oromia, start/finish ambient temperature, start/finish ambient humidity (Colemeter thermometer hygrometer humidity meter).

After 24 h, *M. sorbens* adhering to the sticky trap lids were identified according to taxonomic keys [18,19] and counted. In The Gambian study, *M. sorbens* were counted, females dissected for gravidity (carrying fully developed eggs), and *Musca domestica* were counted. In the Ethiopian study, *M. sorbens* were counted and sexed. Negative binomial regression, which accounted for the over-dispersed nature of trap catch data, was used to model the relationship between trap bait and the number of flies caught. The effect of trap bait on the likelihood that a trapped *M. sorbens* was female was analysed using logistic regression (analyses performed in Stata [v. 15, StatCorp], datasets available in S1 Data).

## Chemistry of bait attraction

**Collection of volatiles.**   Air entrainment of faeces samples was performed only during the Gambian study. Five human faeces samples were collected (50 g) from Farafenni and Boiram into individual sterile polyethyleneterephthalate (PET) cooking bags (Sainsbury's Ltd, UK). Volatiles from the faeces were collected using a portable air entrainment kit (Barry Pye, Rothamsted Research, Harpenden). The bag containing the sample was sealed to an aluminium disc with air inlet and outlet holes using bulldog clips. The air entrainment kit, comprising an inflow and outflow pump and charcoal filters (VWR Chemicals BDH, 10–14 mesh, 50 g, preconditioned under a stream of nitrogen at 150 ˚C for a minimum of two hours), was connected to the bag and air inflow (16 L/min) set higher than outflow (2 x 7 L/min), creating positive pressure to prevent entrance of environmental volatiles. PTFE tubing and rubber ferrules were used for all connections. The apparatus was cleaned before and after use with ethanol (100%, Sigma-Aldrich, Gillingham, UK). Volatiles were collected in the outflow onto Porapak Q polymer (50 mg, mesh size 50/80, Supelco), contained inside a glass tube and held with two plugs of sterile silicanised glass wool ('Porapak tubes'). These had been conditioned prior to use by repeated washing with redistilled diethyl ether and heating to 132 ˚C for 2 h under a stream of constant (charcoal-filtered) nitrogen. After 12 h of volatile collection, Porapak tubes were sealed in ampoules under filtered nitrogen for transport and storage. Ampoules were initially stored at the study site at 4 ˚C (for a maximum of six weeks), then in the UK volatiles were eluted from the Porapak tubes using freshly-distilled diethyl ether ('extract') and stored in vials at -20 ˚C until analysis.

**Fly rearing.**   After counting and identification, live female *M. sorbens* were collected from the sticky traps with blunt forceps and placed into insect rearing cages (BugDorm, 32.5 x 32.5 x

32.5 cm) containing water (soaked into blue paper towelling) and white sugar cubes. Either human faeces (20 g presented on a small pot of damp soil to replicate the 'naturalistic' setting), or full fat Ultra-high temperature processing (UHT) milk (soaked into cotton wool) were given as an oviposition medium and protein source [20]. Every third day larvae were transferred to a larval diet medium: molasses sugar (8 g), dried yeast (7 g), full fat UHT milk (100 mL), water (200 mL) and wheat bran (added until the correct consistency was achieved). Cages containing artificial diet were kept indoors in The Gambia at 25–28 °C and 25–50% relative humidity (RH), while cages containing faeces were kept in natural daylight in a ventilated outbuilding, at 17–44 °C and 8–65% RH, with a roughly 12:12 light/dark photo-period. Six days after they were placed in the cages, two samples of artificial larval diet and one sample of faeces were removed from oviposition cages and the pupariae carefully exposed. Emerging adults were used for GC-EAG; rearing flies on artificial media was not sustainable over more than three generations.

**Coupled gas chromatography-electroantennography (GC-EAG).** *Musca sorbens* females from the laboratory-reared colony were chilled on ice until movement ceased. The heads severed to kill the fly, and abdomens dissected to determine gravidity. One antenna was removed to reduce noise in the recording. The tip of the other antenna was cut off before inserting the antenna into a glass electrode containing a silver/silver chloride wire and filled with Ringer's solution (7.55 g NaCl, 0.64 g KCl, 0.22 g CaCL$_2$, 1.73 g MgCl$_2$, 0.86 g Na$_2$HCO$_3$, 0.61 g Na$_3$PO$_4$/ L distilled water). Another electrode was inserted into the back of the flies' head to complete the circuit. This assemblage was held in a constant stream (1 L/min) of humidified, charcoal-filtered air. Air entrainment extracts were concentrated (from 750 to 50 μL) under a stream of charcoal-filtered nitrogen. A sample of extract (1 μL of a representative 50 μL concentrated 12-h air entrainment extract) was injected onto a 30 m non-polar polydimethylsiloxane (HP1) column (internal diameter 0.32 mm, solid phase thickness 0.52 μm) in a gas chromatograph (Hewlett Packard HP6890 with a cool-on-column injector, hydrogen carrier gas and flame ionisation detector). The following GC program was used: oven temperature maintained at 40 ˚C for 2 min, increased by 5 ˚C per min to 100 ˚C, then raised by 10 ˚C per min to 250 ˚C. Emerging compounds were delivered simultaneously to the flame ionisation detector and the airstream blowing over the antenna. The signal was amplified 10,000 times by the Intelligent Data Acquisition Controller-4, and signals were analysed by using EAD 2000 software (both Syntech, The Netherlands). Antennal responses were correlated visually to compound peaks by overlaying traces on a light box, and the procedure repeated for four female *M. sorbens* flies.

**Coupled gas chromatography-mass spectrometry (GC-MS).** Compounds found to be electrophysiologically-active were identified by GC-MS only. One concentrated extract was diluted tenfold prior to injecting (1 μL) onto a HP1 column (dimensions as for GC-EAG; Hewlett HP 5890 GC fitted with a cool-on-column injector, helium carrier gas and FID, with a deactivated HP1 pre-column [0.53 mm ID]). The following program was used: oven temperature maintained at 30 ˚C for 5 min, increased by 5 ˚C per min to 250 ˚C. A VG Autospec double-focusing magnetic sector mass spectrometer (MS) using electron impact ionisation (70 eV, 250 ˚C) was coupled to the GC and data analysed using an integrated data system (Fisons Instruments, Manchester, UK). Compounds were identified using the NIST 2005 database of standards (NIST/EPA/NIH mass spectral library version 2.0, Office of the Standard Reference Data Base, National Institute of Standards and Technology, Gaithersburg, Maryland).

## Results

### Trapping

In Boiram, a total of 1734 muscid flies were caught across all traps between 17[th] July 2009 and 7[th] August 2009. Of these, 382 were *M. sorbens*, 1046 *M. domestica* and 306 were unidentified

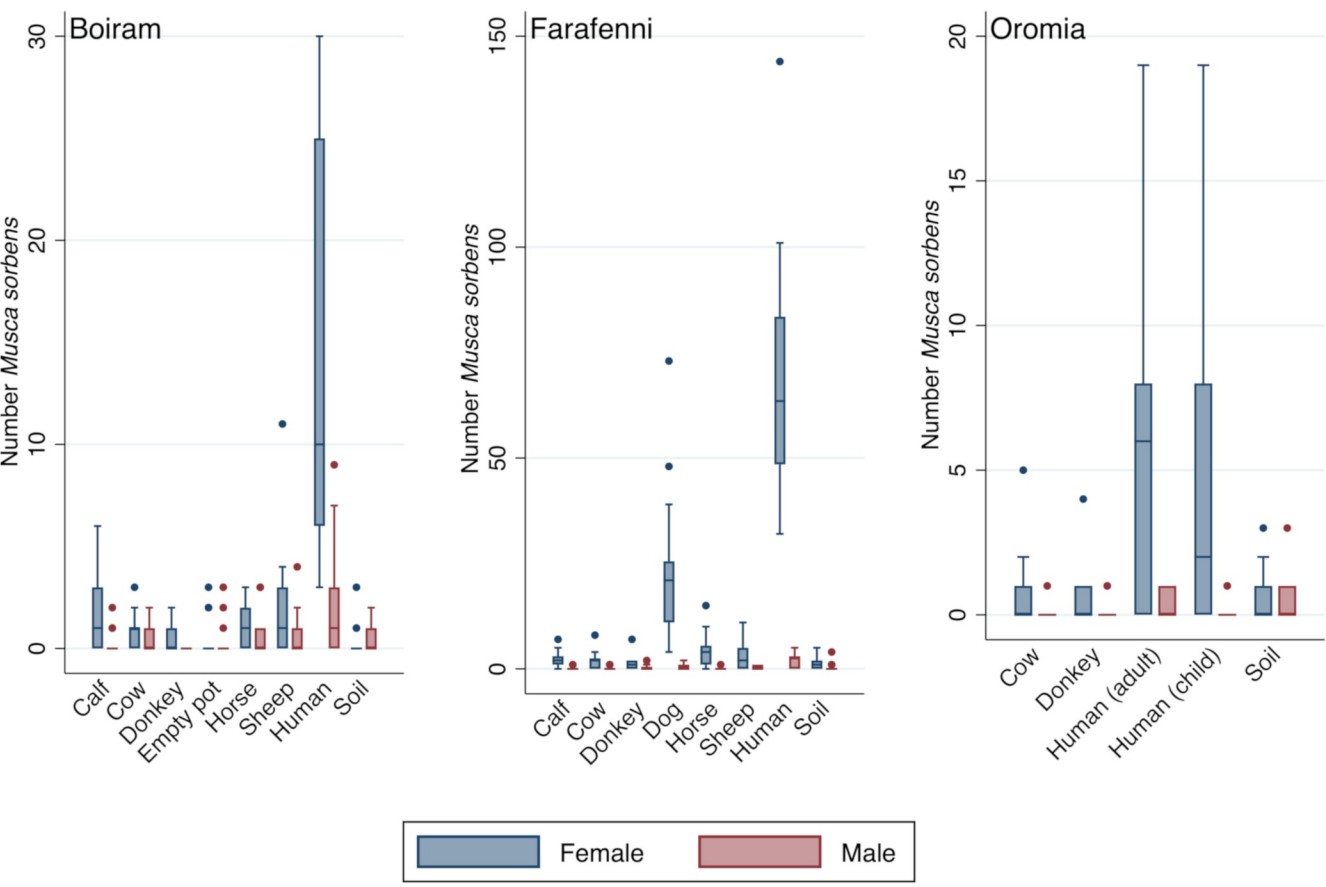

**Fig 2. Male and female *Musca sorbens* caught by different faeces bait, in the three studies (boxes, median and interquartile range; points, outliers).** Trap days per bait are Boiram, *n* = 16; Farafenni, *n* = 16 (sheep *n* = 15); Oromia, *n* = 11.

*Musca* spp. All had a female majority (*M. sorbens*, 82.9%; *M. domestica*, 69.7%; unidentified *Musca* spp., 80.3%). In Farafenni, a total of 1899 flies were caught between 18[th] November 2009 and 10[th] December 2009, of which 1754 were *M. sorbens* and 145 *M. domestica*. Again, the majority of caught flies were female (*M. sorbens*, 96.5%; *M. domestica*, 86.2%). In Boiram and Farafenni, across all faeces bait types other than horse and sheep, more than 60% of females caught were gravid. In Oromia, a total of 152 *M. sorbens* were caught between 19[th] and 27[th] February 2017. Of those that could be sexed (96.1%), most were female (90.4%). Gravidity was not measured, and other flies/arthropods were not counted. The distribution of *M. sorbens* trap catches per day (24-hour period) by bait type are shown in Fig 2, S1 Table, S2 Table.

In all three studies, the number of female *M. sorbens* caught was strongly associated with the type of trap bait used (*P*<0.001, Table 1), and human faeces was consistently and strongly the most attractive bait overall. When comparing female *M. sorbens* trap catch to the soil bait (as the baseline), however, the relative performance of different bait types (estimated rate ratios) varied substantially between studies. In Boiram, 44.4 times more female *M. sorbens* were caught in human faeces-baited traps (mean rate ratio [RR]; 95% confidence intervals [CI] 15.5–127.3, *P*<0.001), in Farafenni 61.4 times more (95% CI 32.3–117.0, *P*<0.001), and in Oromia 10.5 times more in traps baited with adult faeces (95% CI 2.5–43.5, *P* = 0.001) and 8.2 times more in traps baited with child faeces (95% CI 2.0–34.0, *P* = 0.004) (Table 1). Where measured, the strong preference for human faeces was not observed in *M. domestica* (S1 Fig).

For non-human faeces baits, in Farafenni, only dog, horse and sheep faeces caught more female *M. sorbens* than soil-baited traps (dog, RR 20.7, 95% CI 10.8–39.6, *P*<0.001; horse, RR 3.7, 95% CI 1.9–7.3, *P*<0.001; sheep, RR 2.5, 95% CI 1.2–5.1, *P* = 0.012 Table 1). In Boiram, only traps baited with horse, sheep and calf faeces were more attractive than soil-baited traps (RR 3.2, 95% CI 1.0–10.2, *P* = 0.048; RR 5.2, 95% CI 1.7–15.8, *P* = 0.004; RR 4.6, 95% CI 1.5–14.1, *P* = 0.008 respectively, Table 1). Dog faeces were not tested in Boiram nor Oromia. In Oromia no evidence was found of a difference in catch rates between the non-human faeces baits (cow or donkey) and the soil bait (*P*>0.05).

Human faeces were not only more attractive than soil bait, but were found to be more attractive than the second most attractive bait at each site (*P*<0.001). In Boiram, human faeces-baited traps caught 8.5 times as many female *M. sorbens* (RR; 95% CI 4.2–17.2) than sheep faeces-baited traps. In Farafenni, human faeces-baited traps caught 3.0 times more female *M. sorbens* (RR; 95% CI 1.9–4.7) compared to dog faeces-baited traps, and in Oromia, 7.9 (RR; 95% CI 2.0–30.8) and 6.1 (RR; 95% CI 1.6–24.1) times more were caught by human and child faeces-baited traps than cow faeces-baited traps.

In all studies, there were increased odds that *M. sorbens* caught by human faeces bait relative to those caught by soil bait would be female (Boiram, odds ratio [OR] 7.93 [95% CI 2.16–29.1] *P* = 0.002; Farafenni, OR 11.52 [95% CI 4.29–30.96] *P*<0.001; Oromia adult, OR 14 [95% CI 3.63–53.99) *P*<0.001; Oromia child, OR 32.67 [95% CI 5.59–191.06] *P*<0.001; S3 Table). Similarly, increased odds for female *M. sorbens* being caught were observed for dog and horse faeces baits in Farafenni, and cow faeces bait in Oromia (S3 Table). Generally, fewer male *M. sorbens* were trapped than females. In Boiram and Farafenni, only human faeces attracted more males than the soil control (*P*<0.05); this was not observed in Oromia (*P*>0.05, Table 1). Although it appears that males were possibly more affected by climatic variables (average ambient temperature affecting trap catch at each study site, *P*<0.05; ambient relative humidity affecting male catch at Oromia, *P*<0.05 [not measured at other sites], Table 1), low male catch rates may have affected these estimates (total males caught: Bioram, 65; Farafenni, 61; Oromia, 14).

## Coupled gas chromatography-electroantennography

Twelve compounds from the headspace of the human faecal sample elicited an antennal response from two or more female *M. sorbens*. These compounds were subsequently tentatively identified by GC-MS as 3-ethylpentane, 2-methylpropanoic acid, butanoic acid, pentanoic acid, hexanoic acid, cresol, 2-phenylethanol, valerolactam, dimethyl tetrasulphide, indole, 2-dodecanone and an unidentified cholesterol derivative (Table 2).

## Discussion

We found evidence that *M. sorbens*, the putative vector of trachoma, is strongly attracted to odours produced by human faeces, which attracted and caught the greatest number of *M. sorbens* in all three studies. This was particularly so with female *M. sorbens* relative to males. In the Gambian studies, the majority of females were found to be gravid. These results corroborate previous studies finding that female *M. sorbens* are attracted to faeces for oviposition [2,15,21]. They further demonstrate that volatile cues alone are responsible for this attraction, as the faeces baits were only visible to flies if they were very close, directly above, and could see down through the mesh. A previous study conducted in The Gambia [16] demonstrated attraction to human faeces, but because the faeces were not hidden from the flies, visual cues could not be ruled out as a stimulus. Traps with no faeces bait (soil control and empty pot) consistently caught very few flies, suggesting that the volatiles alone were responsible for

**Table 1. Female and male *Musca sorbens* caught by different baits at three study sites in The Gambia (Boiram and Farafenni) and Ethiopia (Oromia).** Soil and/or empty pots were used as negative controls, and the impact of climate (ambient temperature, rainfall/relative humidity) on trap catch assessed.

| | | Female *Musca sorbens* | | | Male *Musca sorbens* | | |
|---|---|---|---|---|---|---|---|
| **BOIRAM** | | Rate ratio (95% CI) | *P*-value[A] | *P*-value[B] | Rate ratio (95% CI) | *P*-value[A] | *P*-value[B] |
| Bait | Calf faeces | 4.60 (1.50–14.12) | 0.008 | | 0.80 (0.61–4.13) | 0.790 | |
| | Cow faeces | 2.20 (0.66–7.31) | 0.198 | | 1.00 (0.21–4.86) | 1.000 | |
| | Donkey faeces | 1.40 (0.39–5.04) | 0.607 | | [C] | | |
| | Empty pot | 1.40 (0.39–5.04) | 0.607 | <0.001 | 1.20 (0.26–5.60) | 0.817 | 0.086 |
| | Horse faeces | 3.20 (1.01–10.15) | 0.048 | | 1.80 (0.41–7.82) | 0.433 | |
| | Human faeces | 44.40 (15.49–127.30) | <0.001 | | 5.60 (1.43–21.96) | 0.014 | |
| | Sheep faeces | 5.20 (1.71–15.83) | 0.004 | | 1.60 (0.36–7.08) | 0.536 | |
| | Soil | 1.0 | | | 1.0 | | |
| Climate | Av. temp[D,E], continuous | 1.15 (0.92–1.45) | | 0.225 | 1.45 (1.08–1.94) | | 0.014 |
| | Rainfall[F,E], Yes | 1.21 (0.40–3.62) | | 0.737 | 0.82 (0.21–3.21) | | 0.779 |
| **FARAFENNI** | | | | | | | |
| Bait | Calf faeces | 2.00 (0.97–4.11) | 0.059 | | 0.33 (0.06–1.78) | 0.198 | |
| | Cow faeces | 1.70 (0.83–3.58) | 0.145 | | 0.50 (0.12–2.17) | 0.355 | |
| | Dog faeces | 20.70 (10.80–39.57) | <0.001 | | 1.17 (0.35–3.85) | 0.800 | |
| | Donkey faeces | 1.20 (0.57–2.63) | 0.607 | <0.001 | 0.83 (0.23–3.00) | 0.781 | <0.001 |
| | Horse faeces | 3.70 (1.85–7.28) | <0.001 | | 1.17 (0.02–1.46) | 0.106 | |
| | Human faeces | 61.40 (32.26–117.02) | <0.001 | | 5.33 (1.97–14.47) | 0.001 | |
| | Sheep faeces | 2.50 (1.22–5.08) | 0.012 | | 0.89 (0.25–3.21) | 0.857 | |
| | Soil | 1.0 | | | 1.0 | | |
| Climate | Av. temp[G], continuous | 1.11 (0.96–1.27) | | 0.150 | 1.26 (1.05–1.52) | | 0.015 |
| | Av. RH[F], continuous | 1.02 (0.98–1.06) | | 0.294 | 1.02 (0.97–1.07) | | 0.412 |
| **OROMIA** | | | | | | | |
| Bait | Cow faeces | 1.33 (0.20–6.35) | 0.718 | | 0.17 (0.02–1.42) | 0.102 | |
| | Donkey faeces | 1.00 (0.21–5.01) | 1.000 | | 0.33 (0.06–1.71) | 0.189 | |
| | Adult human faeces | 10.50 (2.54–43.48) | 0.001 | | 0.50 (0.12–2.09) | 0.342 | |
| | Child faeces | 8.17 (1.96–34.03) | 0.004 | <0.001 | 0.33 (0.06–1.71) | 0.189 | 0.359 |
| | Soil | 1.0 | | | 1.0 | | |
| Climate | Av. temp[G], continuous | 0.89 (0.70–1.14) | | 0.366 | 1.48 (1.08–2.04) | | 0.015 |
| | Av. RH[F], continuous | 1.01 (0.96–1.07) | | 0.721 | 1.09 (1.02–1.16) | | 0.007 |

[A] *P*-value comparing this category with baseline (soil)

[B] *P*-value testing hypothesis that bait is associated with number of flies trapped

[C] Value could not be estimated as no male *Musca sorbens* caught in traps baited with donkey faeces

[D] Adjusted for rainfall

[E] Based on 13 days data (two days missing)

[F] Adjusted for average temperature

[G] Adjusted for average RH

(faeces-baited) trap attractiveness, and that neither adjacent traps, nor nearby faeces sources, influenced trap capture. In the current study, there was also evidence that some of the non-human faeces baits were relatively more attractive than others (i.e. calf and sheep in Boiram attracted more female *M. sorbens* than the soil control, as did dog and horse in Farafenni). The high rate ratio for female *M. sorbens* caught using dog faeces in Farafenni (the only site with dogs) may indicate a preference for the faeces of non-herbivores. The mean rate ratio of *M. sorbens* females caught using human faeces relative to the soil control varied between the three sites, but in all cases a large effect was seen. The Gambian studies were conducted in two

**Table 2. Twelve compounds in the odour of a human faeces sample elicited an electrophysiological response from female *Musca sorben*s.** These were tentatively identified by coupled gas chromatography-mass spectrometry.

| Peak number | Compound identified |
| --- | --- |
| 1 | 3-Ethylpentane |
| 2 | 2-Methylpropanoic acid |
| 3 | Butanoic acid |
| 4 | Pentanoic acid |
| 5 | Hexanoic acid |
| 6 | Isomer of cresol |
| 7 | 2-Phenylethanol |
| 8 | Valerolactam |
| 9 | Dimethyl tetrasulphide |
| 10 | Indole |
| 11 | 2-Dodecanone |
| 12 | Cholesterol derivative |

different ecological settings and at different times of the year, making it impossible to account for the differences in fly abundance between sites. At the Oromia site, trap catch may have been removed by birds or other insectivores as no protective wire frames were used. Even more so, the environment (including local abiotic and biotic factors) in Ethiopia is so different to that in The Gambia that comparison of population density between studies, and based on a small sampling window, would not be appropriate. As well as differences between local conditions and sampling timeframes precluding meaningful comparisons of trap catch between study sites, further limitations of this trapping work are as follows. The same faeces types were not available at each study site, therefore the considerable attractiveness of dog faeces observed in Farafenni was not tested at any other site. Faeces freshness was not taken into consideration; although all faeces baits were collected as fresh as possible, freshness would have varied between samples and may account for, or contribute to, differences in attractiveness seen. The volatile compounds (odour) released by the faeces bait would constitute a stronger signal the fresher the faeces sample, with fresh samples most likely eliciting a stronger attraction response from flies. Future studies could estimate the longevity of attractiveness by measuring faecal moisture content as a proxy for freshness.

Differential attraction to different types of faeces could be due to the presence or absence of certain semiochemicals that attract or repel the flies. Electroantennography was used to determine whether there are chemicals in the odour of human faeces that the antennae of *M. sorbens* detect and respond to. Importantly, the reported antennal responses do not inform of the nature of the response, only indicating that the fly detects the chemical. For this reason, EAG studies should be followed by robustly designed behavioural bioassays, able to differentiate between attractive and repellent compounds. Given the significant numbers of female *M. sorbens* caught in the three trapping studies, and the high proportion of gravidity observed where measured, we speculate that some or all of the compounds identified here as EAG-active serve as oviposition attractants to female flies. Of the *M. sorbens* EAG-active compounds tentatively identified here, short chain fatty acids (SCFA, including 2-methyl propanoic acid, butanoic acid, pentanoic acid and hexanoic acid), indoles, cresols and sulphur compounds are known volatile organic compounds in faeces [22–25]. SCFA are produced in the gut by bacterial fermentation of carbohydrates and proteins [26], and are commonly found in vertebrate-associated volatiles, including urine and faeces [22,27–30]. Their detection by host-seeking arthropods is well documented [31–35]. The aromatic compounds identified (cresol,

2-phenylethanol, indole) are likely to be fermentation products of the aromatic amino acids tyrosine, phenylalanine and tryptophan [24]. All GC-EAG replicates were conducted using the same human faeces odour extract, chosen as a 'representative' sample. This does not negate the importance of the compounds identified as eliciting a response. However, it could be that the sample omitted other compounds that are important in the detection of human faeces by _M. sorbens_, or indeed contained compounds that are not commonly present in faecal samples. This could be resolved in future studies by creating an odour 'blend' of extract for testing, combining odour from several human faeces samples.

Several of these faeces-associated volatiles have been described in similar entomological studies involving other higher Dipterans. Nine compounds in pig faeces volatiles elicited antennal response in _M. domestica_; butanoic acid and indole both eliciting strong responses in subsequent dose-response EAG [36]. Both _m_- and _p_-cresol in the headspace of canine faeces elicited antennal response in the green bottle fly, _Lucilia sericata_, and a chemical blend including these compounds was as attractive to flies as the faeces [37]. Cresols (isomer not identified) in volatiles from rat carrion also elicited antennal response from _L. sericata_ [38], and female and male _M. domestica_ responded (by EAG) to volatiles including butanoic acid, hexanoic acid, 2-phenylethanol and _p_-cresol in the headspace of vinegar [39]. Similarly, butanoic acid and _p_-cresol were among chemostimulants of _Stomoxys calcitrans_ detected in the headspace of rumen volatiles, and were also found to elicit activation and attraction in a wind tunnel [40]. Microbial degradation is thought to lead to the production of _m_- and _p_-cresol in cattle urine, again found to elicit an EAG response in _S. calcitrans_ [41]. _Stomxys calcitrans_ is a muscid fly with coprophagous larvae, known to be attracted to faecal odours. This fly has been shown to select faeces by their odour [42], as demonstrated here with _M. sorbens_, and the chemostimulant compounds thought to be responsible for that attraction included butanoic acid, indoles, _p_-cresol and sulphides.

Taken together, the faecal semiochemicals described here are commonly isolated and/or detected because they are products of bacterial decomposition. As such, they are frequently detected by filth flies, most likely as cues for oviposition sites. To underpin the specific attractiveness of human faeces to _M. sorbens_ therefore, either variation in amounts emitted, in the ratios of compounds present, or the presence of further compounds not identified in this study, must distinguish human faeces as a preferable oviposition medium.

## Conclusion

Our study demonstrates that female _M. sorbens_ at three different study locations, in both West Africa and East Africa, are preferentially attracted to the volatiles of human faeces, as evidenced by attraction in the absence of visual cues. We provide evidence that twelve compounds are putative attractants that may play a role in this response, by identifying, for the first time, compounds including short chain fatty acids and aromatic compounds that are detected by the antennae of _M. sorbens_. Further work is required to optimise chemical blends and release rates, to produce a synthetic lure to which the behavioural responses of _M. sorbens_ can be investigated. Establishing those with attractive properties may lead to the design of baits for odour-baited traps, which could be used for _M. sorbens_ surveillance or even population suppression or control.

## Supporting information

**S1 Table. Median female _Musca sorbens_/trap/24-hours (IQR).**
(DOCX)

**S2 Table. Median male *Musca sorbens*/trap/24-hours (IQR).**
(DOCX)

**S3 Table. Odds of *Musca sorbens* caught by different bait types being female.**
(DOCX)

**S1 Data. *Musca sorbens* trapping dataset (raw).**
(XLS)

**S1 Fig. Male and female *Musca domestica* caught by different faeces bait, in the two studies in The Gambia (boxes, median and interquartile range; points, outliers).** Trap days per bait are Boiram, *n* = 16; Farafenni, *n* = 16 (sheep *n* = 15).
(TIF)

## Acknowledgments

The authors thank all of the families and communities at the study sites for allowing us to conduct this work around their homes. We are particularly grateful for the contributions of the MRC Gambia fieldworkers Pateh Makalo, Kemo Ceesay and Dembo Camara, without whose hard work, dedication and ingenuity this study would not have been possible, and for the assistance of Keol Debele, the entomology fieldworker in Oromia. We are also grateful for the assistance from Aida Abashawl in project co-ordination, Nazif Jemal in communications, and Munira Haji Mohammed Yusuf, Helen W/Semayat, and Demitu Legesse in translation in the Oromian work.

## Author Contributions

**Conceptualization:** Robin L. Bailey, John A. Pickett, Sarah Dewhirst, Vanessa Chen-Hussey, Anna Last, Matthew J. Burton, Steve W. Lindsay, James G. Logan.

**Data curation:** Julie Bristow.

**Formal analysis:** Ailie Robinson, Julie Bristow, David Macleod.

**Funding acquisition:** Robin L. Bailey, Virginia Sarah, Anna Last, Matthew J. Burton, Steve W. Lindsay, James G. Logan.

**Investigation:** Julie Bristow, Matthew V. Holl, Pateh Makalo, Michael A. Birkett, John C. Caulfield, John A. Pickett, Christine M. Woodcock.

**Methodology:** Julie Bristow, Michael A. Birkett, John C. Caulfield, John A. Pickett, Sarah Dewhirst, Vanessa Chen-Hussey, Steve W. Lindsay, James G. Logan.

**Project administration:** Julie Bristow, Matthew V. Holl, Wondu Alemayehu, Virginia Sarah, Sarah Dewhirst, Vanessa Chen-Hussey, Anna Last, James G. Logan.

**Resources:** Wondu Alemayehu, Michael A. Birkett, Virginia Sarah, Umberto D'Alessandro.

**Supervision:** Michael A. Birkett, John C. Caulfield, John A. Pickett, Sarah Dewhirst, Vanessa Chen-Hussey, Umberto D'Alessandro, Anna Last, Matthew J. Burton, Steve W. Lindsay, James G. Logan.

**Visualization:** Ailie Robinson, Julie Bristow, David Macleod.

**Writing – review & editing:** Christine M. Woodcock.

**Writing – original draft:** Ailie Robinson, Julie Bristow, James G. Logan.

**Writing – review & editing:** Ailie Robinson, Julie Bristow, Matthew V. Holl, Pateh Makalo, Wondu Alemayehu, Robin L. Bailey, David Macleod, John C. Caulfield, Virginia Sarah, John A. Pickett, Sarah Dewhirst, Vanessa Chen-Hussey, Umberto D'Alessandro, Anna Last, Matthew J. Burton, Steve W. Lindsay, James G. Logan.

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
