## [Decision Letter · Decision Letter 0]

23 Nov 2019

Dear Dr Robinson:

Thank you very much for submitting your manuscript "Responses of the putative trachoma vector, Musca sorbens, to volatile semiochemicals from human faeces" (PNTD-D-19-01361) for review by PLOS Neglected Tropical Diseases. Your manuscript was fully evaluated at the editorial level and by independent peer reviewers. The reviewers appreciated the attention to an important topic but identified some aspects of the manuscript that should be improved.

We therefore ask you to modify the manuscript according to the review recommendations before we can consider your manuscript for acceptance. Your revisions should address the specific points made by each reviewer.

(1) A letter containing a detailed list of your responses to the review comments and a description of the changes you have made in the manuscript.

(2) Two versions of the manuscript: one with either highlights or tracked changes denoting where the text has been changed (uploaded as a "Revised Article with Changes Highlighted" file ); the other a clean version (uploaded as the article file).

(3) If available, a striking still image (a new image if one is available or an existing one from within your manuscript). If your manuscript is accepted for publication, this image may be featured on our website. Images should ideally be high resolution, eye-catching, single panel images; where one is available, please use 'add file' at the time of resubmission and select 'striking image' as the file type. 

Please provide a short caption, including credits, uploaded as a separate "Other" file. If your image is from someone other than yourself, please ensure that the artist has read and agreed to the terms and conditions of the Creative Commons Attribution License at http://journals.plos.org/plosntds/s/content-license (NOTE: we cannot publish copyrighted images). 

(4) Appropriate Figure Files 

Please remove all name and figure # text from your figure files upon submitting your revision. Please also take this time to check that your figures are of high resolution, which will improve both the editorial review process and help expedite your manuscript's publication should it be accepted. Please note that figures must have been originally created at 300dpi or higher. Do not manually increase the resolution of your files. For instructions on how to properly obtain high quality images, please review our Figure Guidelines, with examples at: http://journals.plos.org/plosntds/s/figures

While revising your submission, please upload your figure files to the Preflight Analysis and Conversion Engine (PACE) digital diagnostic tool, https://pacev2.apexcovantage.com/ PACE helps ensure that figures meet PLOS requirements. To use PACE, you must first register as a user. Then, login and navigate to the UPLOAD tab, where you will find detailed instructions on how to use the tool. If you encounter any issues or have any questions when using PACE, please email us at figures@plos.org.

We hope to receive your revised manuscript by Jan 22 2020 11:59PM. If you anticipate any delay in its return, we ask that you let us know the expected resubmission date by replying to this email.

To submit your revised files, please log in to https://www.editorialmanager.com/pntd/

Sincerely,

Jeremiah M. Ngondi, MB.ChB, MPhil, MFPH, Ph.D

Associate Editor

Mathieu Picardeau

Deputy Editor

Reviewer's Responses to Questions

**Key Review Criteria Required for Acceptance?**

**Methods**

-Are the objectives of the study clearly articulated with a clear testable hypothesis stated?

-Is the study design appropriate to address the stated objectives?

-Is the population clearly described and appropriate for the hypothesis being tested?

-Is the sample size sufficient to ensure adequate power to address the hypothesis being tested?

-Were correct statistical analysis used to support conclusions?

-Are there concerns about ethical or regulatory requirements being met?

Reviewer #1: Objectives could have been more strongly stated. A trapping system was developed to capture adult flies and flies were reared to produce adults for the GC-EAG studies. Description of traps is confusing. One complete trap design should be described, then the modifications made to arrive at the next design and so on. Line 165 - sounds like the yellow sticky trap is really a yellow disc coated with adhesive, but do not know. Lines 166-167 - Nylon and polyester mesh used to prevent entry of flies into the traps, correct? Is there a mesh size to be used? Line 170 - which company manufacturers Rat Stop? Lines 172-173 - Wire frames. Made from individual wires, pieces of wire fence or what? Coated in black electrical tape? Size of frames given in cubic cm. This can be many shapes. Need length, width and height. Discussion of feces collections do not strongly state how fresh the feces must be to qualify for the studies. Line 196 - soil from uncontaminated areas. Contaminated with what? Collected near trap sites. How near or far? Moisture content of feces would be interesting to know. Line 201 - traps were separated by 50 cm. This seems a bit close. Line 227 - was the portable air entrainment kit made by the authors, or a commercial model? Fly rearing paragraph could be more clearly stated. Line 252 - need dimensions of BugDorms, not cubic cm. How many BugDorms and how many flies in each? Line 253 - what is blue roll? Lines 253-254 - on a small pot of damp soil? Lines 264-266 - run on sentence. Actual photos of the traps would be better than the casual drawings. Was GC-EAG done on only 4 flies?

Reviewer #2: (No Response)

Reviewer #3: Well done authors!

This is an important piece of work which is well over-due. The objectives and methods are clearly stated and described and the interpretation appropriate. This represents a HUGE amount of work!!

I have a couple of minor comments and noticed a few typos

Major comments:

None

Minor comments:

1) In the title and throughout the ms the authors use the term "putative trachoma vector". I suggest removing the word 'putative' as there is no longer any reasonable question as to whether M. sorbens is a trachoma vector. M. sorbens is a mechanical vector of trachoma, it does not however, have an obligatory role in the transmission of trachoma as Aedes mosquitoes do in the transmission of human Yellow Fever and Dengue. It would be worth mentioning this in the introduction. The relative importance of eye-seeking flies in trachoma transmission will likely vary in time and space. Controlling eye-seeking flies will not block trachoma transmission. Stopping fly-eye contact will not block trachoma transmission.

2) The findings of the work do not identify the role of the electrophysiologically active semio-chemicals identified. It is reasonable to assume that they are attractants for oviposition, since their presence was associated with a convincingly high proportion of gravid female sorbens. However, the presence of an electrophysiological response does not in itself predict behaviour. The compounds may act individually as attractants or repellents, or in combination elicit a different response. This should be the focus of future work

Typos and other nit-picking:

Lines 127-130: Starting 'It is unknown however, ..." the sentence is presented as an 'either/or' construction, when both are reasonable suggestions. I would suggest constructing the sentence to be 'The more prolific emergence could be a result of better larval survival, more eggs being laid as a result of its greater relative attractiveness to ovipositing flies or other factors'. For me I'd continue to consider it as the preferred and 'better' medium for larval development due to the size of the flies emerging after pupation.

Line 145: Repetition of The Gambia

Line 217: Suggest defining gravidity for the lay-audience and the specific context. It is typically used to mean 'mated with developed eggs', was the stage of gravidity estimated?

Line 300: Any idea what the 'unidentified Musca spp' were in Oromia? That's intriguing. Any idea if these unidentified species are caught from human eyes too?

Line 306: Do you mean 'per night' or per 24hr period?

Line 379: "...a preference for non-herbivore faeces." made me chuckle. Surely the preference is for "the faeces of non-herbivores"?

**Results**

-Does the analysis presented match the analysis plan?

-Are the results clearly and completely presented?

-Are the figures (Tables, Images) of sufficient quality for clarity?

Reviewer #1: Results are weakly stated with relation to the stats. Lines 302-303 - For feces baits other than horse and sheep, > 50% of flies collected were gravid. Lines 303-305 - run on sentence. Too long. Figure 2 might be more effective in tabular form with means and standard deviation show along with letters indicating statistical separation of means. Tables need more complete titles.

Reviewer #2: (No Response)

Reviewer #3: (No Response)

**Conclusions**

-Are the conclusions supported by the data presented?

-Are the limitations of analysis clearly described?

-Do the authors discuss how these data can be helpful to advance our understanding of the topic under study?

-Is public health relevance addressed?

Reviewer #1: Data basically support conclusions. Limitations of analysis not mentioned. Authors discuss how data might be helpful. Authors do not mention the possibility of feces management. This might be difficult to impossible, but people need to know that these are sources of the flies that are transmitting disease to them and their children. Lines 384-387 and lines 405-408 - run on sentences. Too long.

Reviewer #2: (No Response)

Reviewer #3: (No Response)

**Editorial and Data Presentation Modifications?**

Reviewer #1: The paper could use a thorough revision for clarity and organization. I think the science is basically ok, but the paper can be difficult to read in some places. I have tried to attach a scan of the paper so more of my comments can be seen. I would call this revision minor.

Reviewer #2: (No Response)

Reviewer #3: (No Response)

**Summary and General Comments**

Reviewer #1: The study is a very simple one, but very important for determining the breeding materials used by this fly. Also the evaluation of chemical compounds found in the feces might be developed into efficacious baits. The sentence structure needs to be tightened up and things more clearly stated.

Reviewer #2: This is an excellent paper overall. The writing is impeccable. With the exception of a single typographical error on line 202 (bais for bias) I did not find any errors throughout the manuscript.

The work is important, and the degree of replication in different study sites sufficient to make the results quite convincing.

A few minor comments for the authors consideration:

1) A photograph of the assembled trap would be helpful, either instead of or supplemental to the schematic drawing.

2) Most of the flies collected from Boiram were M. domestica. Although the focus of the paper was on M. sorbens, it would be interesting to see the collection results for domestica as well. How were the house fly collections distributed? Was the pattern substantially different from that of M. sorbens?

3) The climate information is mentioned in the Methods but never mentioned again except for some entries in Table that I was unable to decode. Suggest removing the climate data or discussing it in the context of the collection data.

4) The number of flies examined by EAG seems rather small. Although I have no expertise with this type of work I wonder whether an EAG response by two flies is sufficient to make conclusions about the attractiveness of the chemical constituents.

Reviewer #3: (No Response)

PLOS authors have the option to publish the peer review history of their article (what does this mean?). If published, this will include your full peer review and any attached files.

Reviewer #1: No

Reviewer #2: No

Reviewer #3: No

---

## [Editor Report · Decision Letter 1]

13 Jan 2020

Dear Dr Robinson,

We are pleased to inform you that your manuscript, "Responses of the putative trachoma vector, Musca sorbens, to volatile semiochemicals from human faeces", has been editorially accepted for publication at PLOS Neglected Tropical Diseases.

Before your manuscript can be formally accepted and sent to production you will need to complete our formatting changes, which you will receive in a follow up email. Please note: your manuscript will not be scheduled for publication until you have made the required changes.

IMPORTANT NOTES

* Copyediting and Author Proofs: To ensure prompt publication, your manuscript will NOT be subject to detailed copyediting and you will NOT receive a typeset proof for review. The corresponding author will have one final opportunity to correct any errors when sent the requests mentioned above. Please review this version of your manuscript for any errors.

* If you or your institution will be preparing press materials for this manuscript, please inform our press team in advance at plosntds@plos.org. If you need to know your paper's publication date for media purposes, you must coordinate with our press team, and your manuscript will remain under a strict press embargo until the publication date and time. PLOS NTDs may choose to issue a press release for your article. If there is anything that the journal should know, please get in touch.

*Now that your manuscript has been provisionally accepted, please log into EM and update your profile. Go to http://www.editorialmanager.com/pntd, log in, and click on the "Update My Information" link at the top of the page. Please update your user information to ensure an efficient production and billing process.

*Note to LaTeX users only - Our staff will ask you to upload a TEX file in addition to the PDF before the paper can be sent to typesetting, so please carefully review our Latex Guidelines [http://www.plosntds.org/static/latexGuidelines.action] in the meantime.

Best regards,

Jeremiah M. Ngondi, MB.ChB, MPhil, MFPH, Ph.D

Associate Editor

Mathieu Picardeau

Deputy Editor

---

## [Editor Report · Acceptance letter]

27 Feb 2020

Dear Dr Robinson,

We are delighted to inform you that your manuscript, "Responses of the putative trachoma vector, *Musca sorbens*, to volatile semiochemicals from human faeces," has been formally accepted for publication in PLOS Neglected Tropical Diseases.

Best regards,

Serap Aksoy

Editor-in-Chief

Shaden Kamhawi

Editor-in-Chief
